# The Chemically Modified Leaves of *Pteris vittata* as Efficient Adsorbent for Zinc (II) Removal from Aqueous Solution

Qaiser Khan [1], Muhammad Zahoor [2,*], Syed Muhammad Salman [1], Muhammad Wahab [1], Muhammad Talha [2], Abdul Waheed Kamran [3], Yousaf Khan [3], Riaz Ullah [4], Essam A. Ali [5] and Abdul Bari Shah [6]

1 Department of Chemistry, Islamia College University, Peshawar 25000, Khyber Pakhtunkhwa, Pakistan
2 Department of Biochemistry, University of Malakand,
Chakdara Dir Lower 18800, Khyber Pakhtunkhwa, Pakistan
3 Department of Chemistry, University of Malakand,
Chakdara Dir Lower 18800, Khyber Pakhtunkhwa, Pakistan
4 Department of Pharmacognosy, College of Pharmacy, King Saud University, Riyadh 11451, Saudi Arabia
5 Department of Pharmaceutical Chemistry, College of Pharmacy, King Saud University,
Riyadh 11451, Saudi Arabia
6 Division of Applied Life Science (BK21 Plus), IALS, Gyeongsang National University,
Jinju 52828, Republic of Korea
* Correspondence: mohammadzahoorus@yahoo.com

**Abstract:** High concentrations of zinc along with other metals are released by steel mills, and this has a number of negative effects on organism health; most notably, neurological symptoms have been recorded with a high risk of brain atrophy. In the current study, Zn (II) was eliminated from steel mill effluent, utilizing chemically processed *Pteris vittata* plant leaves as a biosorbent. Fourier transform infrared spectroscopy (FTIR), scanning electron microscopy (SEM), thermal gravimetric analysis (TGA), and energy dispersive X-ray spectroscopy (EDX) were applied to characterize the chemically modified *Pteris vittata* leaves, from now onward abbreviated as CMPVL. In order to identify the ideal parameter, batch studies were conducted varying a single parameter affecting the biosorption process at a time, including variations in temperature (293–323 K), initial metal concentration (20–300 mg/L), and adsorbent doses (0.01–0.12 g), pH (2–8), as well as contact time (10–140 min). To describe the isothermal experimental results, a number of models were used including Freundlich, Langmuir, Temkin, Jovanovich, and Harkins–Jura. Among these models, the Langmuir model provided a significant fit to the isotherm data with an $R^2$ of 0.9738. The kinetics data were fitted to the pseudo first order, pseudo second order, power function, Natarajan–Khalaf, and intraparticle diffusion models. The highest $R^2$ (0.9976) value was recorded for the pseudo second order model. Using the Langmuir isotherm, the highest uptake ability (84.74 mg/g) of Zn was recorded. The thermodynamic investigation, carried out at various temperatures, led to the conclusion that the biosorption process was exothermic and spontaneous in nature. The CMPVL, thus, has the potential to function well as an alternative to existing carbon-based adsorbents in the effective elimination of zinc from aquatic environments.

**Keywords:** aquatic environment; adsorption; pollutants; zinc

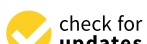



## 1. Introduction

Heavy metals being released into the surroundings as a result of industrial activity is a serious global issue, which directly or indirectly affects human health. The heavy metals released can then be mobilized in soil and, in turn, taken up by plants that are consumed by human populations. Thus, the entry of heavy metals into both soil and water is a major threat towards the destabilization of the ecosystem and harmful to organisms including humans when it enters their bodies through the food chain. In numerous regions of the world, recent rapid industrialization and urbanization have resulted in causing havoc with

human health due to heavy metal acute toxicity [1–3]. Metals are non-biodegradable, and beyond their permissible level, they are very toxic and hazardous [4–6]. When present in trace amounts, zinc serves as a micronutrient and is thought to be vital to life. Zinc, which is present in small amounts in humans (around 2 g), is crucial for the production of proteins as well as DNA polymerization. However, taking in an excessive amount of zinc is hazardous. The symptoms of zinc poisoning include wrath, rigidity of the muscles, nausea, and a lack of appetite [7]. Both natural water and some industrial effluents contain this hazardous metal in excess quantities that needs to be reclaimed [8]. Moreover, inhaling Zn emissions can cause "Zinc Fever," which is marked by cold and a fever, and sometimes, zinc chloride ($ZnCl_2$) smoke vapor-induced lung edema that can be fatal in most instances. Internal organ damage and even death have resulted from large doses of soluble salts, such as $ZnSO_4$ (around 10 g) [9]. According to reports from the WHO, the permissible value of zinc within potable water is 5 mg/L [10]. The majority of Zn that reaches the environment originates from anthropogenic activities such as mining, producing steel, purifying Zn, burning coal, and so on. Other sources that contribute to the high concentration of zinc in the industrial effluent includes pharmaceuticals, galvanizing, paints, pigments, insecticides, cosmetics, etc. [11]. Heavy metal remediation is employed to minimize the quantity of heavy metals in effluent and drinking water to a safe level. Different approaches are used to remove heavy metals from industrial effluent. For example, membrane separation, ion exchange, adsorption, electro dialysis, precipitation, and reverse osmosis are all utilized for the elimination of heavy metals from the aquatic environment [12,13]. The most efficient, affordable, and simple to use of these treatments is biosorption. Activated carbon is most frequently applied as an adsorbent in the reclamation of industrial effluent. However, the increased price of activated carbon has motivated researchers to look for affordable adsorbents, particularly in third world countries [14–17]. Another method for removing metals from soil and water is phytoremediation, which involves plants. However, it has several limitations, such as the accumulation of heavy metals in their tissues and a slow rate of removal [18]. To effectively use this natural capability of plants, one should convert their biomass to adsorbent as adsorption is the most versatile and efficient method. When combining these two ideas, we can effectively remove heavy metals from industrial effluents.

In the current research work, the idea of phytoremediation was utilized to design an efficient and novel biosorbent. For this purpose, nine wild plants (*Pteris vittata, Populus nigra, Eucalyptus camaldulensis, Persicaria maculosa, Arundodonax, Xanthium strumarium, Verbascum thapsus, Ricinus communis,* and *Parthenium hysterophorus*) growing on the bank of the steel mill Dargai Malakand in Pakistan were taken and tested for their phytoremediation capabilities [15]. Among them, *Pteris vittata* with optimum phytoremediation capability was utilized in designing an adsorbent to utilize the intrinsic properties of plants. The final sink in phytoremediation phenomena are leaves; therefore, its leaves were chemically modified to design an efficient biosorbent for the removal of Zinc from steel mill effluents. The biosorbent abbreviated as CMPVL was characterized by a number of instrumental techniques as per the given details in the Experimental Section and was used as an adsorbent to remediate zinc from effluents of the steel industry located in District Dargai, KPK, Pakistan.

## 2. Experimental Set up

### 2.1. Preparation of Adsorbate Solutions

Zinc sulphate heptahydrate ($ZnSO_4·7H_2O$) was used to prepare 1000 ppm standard stock solution. Working solutions for the batch test at concentrations ranging from 20–300 ppm were prepared from it using the dilution formula.

## 2.2. Preparation of Biosorbent

Zn (II) was found to be present in high levels in the effluent of steel mills operating in Dargai, Pakistan. The potential of several plants to perform phytoremediation was tested; these were taken from the drainage line's bank of the steel mills. Among these plants, because of its remarkable phytoremediation abilities, the *Pteris vittata* plant was chosen [15]. To make biosorbent, the chosen plant's leaves were gathered from the selected locality. These were first cleaned utilizing tap water, followed by distilled water to eliminate the dirt and dissolved pollutants and allowed to drying in the shade. The leaves were heated for a while in an oven set to 50 °C to remove moisture, followed by grinding into fine powder.

## 2.3. Chemical Modification of Biosorbent

Approximately 100 g mass of ground leaf powder was taken and immersed in a 0.1 M $HNO_3$ solution for 24 h until it was filtered with 42 Whatman filter paper as well as frequently rinsed in distilled water. The $HNO_3$ solution was utilized to remove previously trapped metals (Fe, Zn, Cr) from the CMPVL. For neutralization process, 0.1 M NaOH was used. Before being baked in an oven at 100 °C, the neutralized biosorbent was allowed to dry in shade. For activation of biosorbent, the whole acid-treated mass was added in the 0.1 M solution of calcium chloride. After this, the treated biosorbent was desiccated in an oven. The activation process was necessary in order to provide a suitable surface (uniform, cylindrical, shrunken, and ruptured edges) for adsorption [19].

## 2.4. Characterization of CMPVL

Functional groups in the biomass of CMPVL were determined, employing an FTIR spectrophotometer (PerkinElmer, Merlin model 2000). The range of wavelength was from 400 to 4000 $cm^{-1}$ and spectra were recorded both before and after biosorption. The structural characterization of the active and loaded samples of CMPVL was inspected by employing scanning electron microscopy (model JSM 59, JEOL, Tokyo, Japan). The TGA analysis was carried out by employing a thermo gravimetric analyzer (Pyres Diamond Series TG, Perkin Elmer, Chicago, IL, USA), utilizing the temperature up to 800 °C to conclude the degradation of adsorbent with increase in temperature. The elemental compositions of the unloaded as well as loaded adsorbent were examined by EDX (JEOL USA JSM-5910). A Quantachrome surface area analyzer (NOVA, 2200e, Boynton Beach, FL, USA) was used to study surface area and pore parameters of the biosorbent.

## 2.5. Adsorption Experiment

A stock solution (1000 ppm) of Zn (II) was made to conduct the adsorption tests, and using the dilution formula, different concentrations of solutions ranging from 20–300 ppm were prepared from the mentioned stock solution, which was then used in the batch adsorption experiments. For the batch experiment, 0.1 g biomass of CMPVL was stirred with Zn (II) solution (50 mL) for 2 h. To determine the quantity of zinc (II) in the filtrate, the mixture was filtered and subjected to an atomic absorption spectrophotometer. The uptake capacity $q_e$ (mg/g) and percent elimination of Zn (II) was measured applying the given formulae.

$$q_e = (C_i - C_f) \times \frac{V}{m} \tag{1}$$

$$\% R = \frac{(C_i - C_f)}{C_i} \times 100 \tag{2}$$

Here $C_i$ (mg/L) shows initial Zn (II) concentration and $C_f$ (mg/L) final concentration after adsorption. The adsorbate volume (V) is measured in liters, and the adsorbent's mass (m) is quantified in grams.

*2.6. Isotherm Analysis*

Biosorption of Zn (II) on CMPVL was examined in solutions with concentrations ranging from 20–300 mg/L. Other parameters such as pH (6), volume of solution (50 mL), contact time (2 h), and adsorbent dose (0.1 g), were kept constant. Using equation (1), the $q_e$ values were determined and plotted against concentration. Different kinds of isothermal models such as Freundlich, Langmuir, Temkin, Jovanovich, and Harkins–Jura were applied to evaluate the isothermal data.

*2.7. Kinetic Study*

To study the kinetics of Zn (II) biosorption on CMPVL, a definite mass of adsorbent (0.1 g) was introduced to Zn (II) solution (100 mg/L) and stirred at 130 rpm for two hours. The kinetic data of biosorption was subjected to a variety of models such as pseudo first order, pseudo second order, power function, Natarajan–Khalaf models, and intraparticle diffusion.

*2.8. Influence of pH and Adsorbent Dose*

Batch adsorption experiments were conducted to examine the impact of pH and CMPVL dose in the reclamation of Zn (II) from aqueous solution (100 mg/L) keeping all the other conditions constant. The solution's pH was set between 2–8 for the batch experiments using NaOH (0.1 M) and $HNO_3$ (0.1 M). To determine the impact of adsorbent dosages on elimination of Zn (II), various doses (0.01–0.12 g) were used, keeping other parameters constant as mentioned before.

*2.9. Thermodynamic Study*

The influence of temperature on the biosorption of Zn (II) was investigated at varying temperatures (293 K, 303 K, 313 K, and 323 K) keeping other experimental conditions constant.

## 3. Result and Discussion

*3.1. Characterization of CMPVL*

3.1.1. FTIR Spectra of Unloaded and Loaded Samples

Functional groups of both the Zn (II) loaded and unloaded biosorbent were visualized using FTIR spectroscopy. Figure 1A,B shows the spectra of treated and loaded biosorbent. In Figure 1A, the spectra of the treated biosorbent indicate that the $NH_2$ group is evident from the peak at 3200 to 3400 $cm^{-1}$, whereas the C-H stretch is evident from the peak at 3000 to 3100 $cm^{-1}$. Carbonyl group stretching can be seen between 1630 and 1680 $cm^{-1}$, while N-H group stretching at 600 $cm^{-1}$. The C-N group stretching is apparent in the spectra between 1100 and 1300 $cm^{-1}$ [20]. The loaded biosorbent showed that N-H stretching is apparent at 3300 $cm^{-1}$ and C-H stretching is there at 2920 $cm^{-1}$. The peak at 1630 $cm^{-1}$ is for C=O stretching or for N-H bending as perhaps both may overlap to give a single peak at 1030 $cm^{-1}$ that may also represent C-H stretching. The decreased band intensities in the spectra of the Zn loaded sample show that the functional groups of the treated biosorbent were occupied with Zn (II), thus demonstrating that Zn was adsorbed.

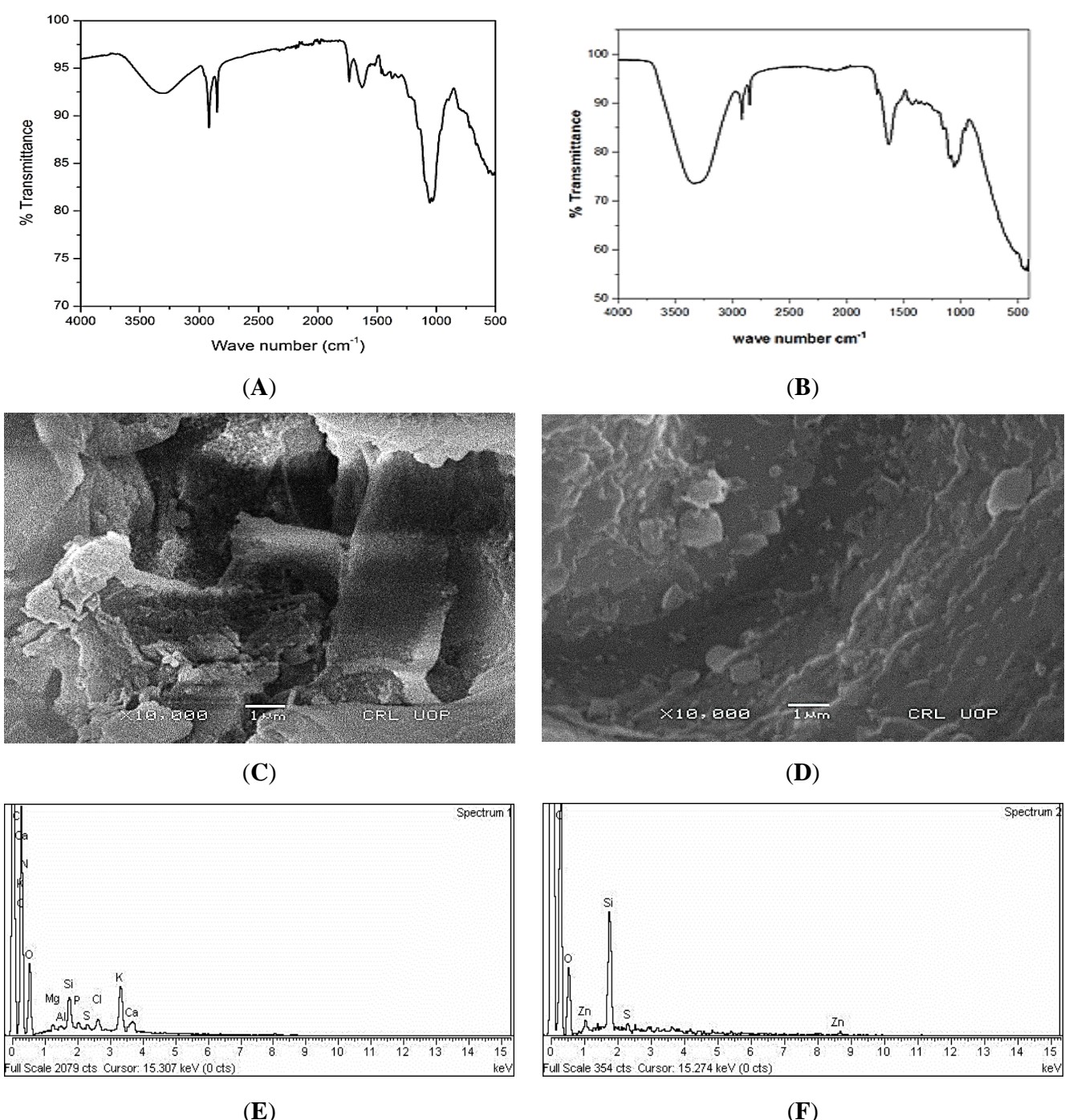

**Figure 1.** *Cont.*

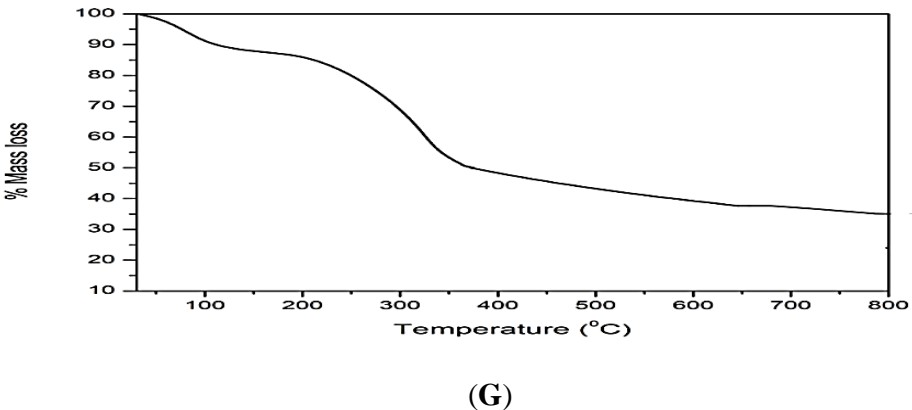

**(G)**

**Figure 1.** Characterization of biosorbent (**A**) FTIR spectra of CMPVL, (**B**) Zn loaded FTIR spectra of CMPVL, (**C**,**D**) SEM images of CMPVL and Zn loaded CMPVL, (**E**) EDX spectra of CMPVL, (**F**) EDX spectra of Zn loaded CMPVL, (**G**) TGA of CMPVL.

### 3.1.2. SEM Study

SEM was employed to examine the biosorbent at a 10,000× magnification. The chemically treated and Zn-loaded biosorbent SEM images are shown in Figure 1C,D. These images clarify the surface morphology and texture of the unloaded and Zn (II) loaded biosorbent. The SEM images depict porous structures with curved surfaces and bent edges that are ideal for use as a biosorbent [21]. The treated biosorbent was seen to have firmness, cylinder, contraction, and shattering of the structure. As shown in the SEM images, after adsorption, there are very few changes as zinc ions are very small and it is difficult to observe the adsorbed atoms with SEM on the biosorbent surface.

### 3.1.3. EDX Study

The elemental study of the unloaded and loaded adsorbent is depicted in Figure 1E,F. The chemically treated adsorbent presented more distinct peaks of oxygen and carbon, whereas small peaks of Mg, K, Ca, Cl, Si, and P were also observable, which may have appeared due to impurity. After biosorption of Zn (II) on CMPVL, the loaded biosorbent presented an evident peak of Zn, confirming the biosorption of Zn.

### 3.1.4. TGA Analysis

The TGA chart is presented in Figure 1G. In thermal gravimetric analysis, a treated adsorbent sample weighing 7.873 mg was used. As per the increase in temperature, the mass loss was observed in three stages. Water evaporation accounted for the first stage of mass loss in relation to temperature, whereas cellulose breakdown contributed to the second mass loss. In the third step, the sample stabilized at a temperature of around 800 °C, resulting in the formation of carbonaceous material [21].

### 3.1.5. Surface Area and Pore Volume

Table 1 displays the CMPVL's surface area and pore volume. The data indicate that CMPVL is a more favorable biosorbent in terms of biosorption capability due to its greater surface area and better pores [21].

**Table 1.** Surface parameters of adsorbent.

| Biosorbent | CMPVL |
|---|---|
| BET surface area (m$^2$/g) | 73.280 |
| Pore volume (cc/g) | 0.820 |
| Mesopore volume (cm$^3$/g) | 0.042 |
| Micropore volume (cm$^3$/g) | 0.342 |
| Pore diameter (A°) | 128.59 |

### 3.2. Adsorption Isothermal Investigation

The amount of Zn removed by the CMPVL increased with the initial concentration of Zn (II) until a specific point, which is considered the optimum concentration. Beyond that, no further increases in biosorption capacity were recorded since the biosorbent had attained saturation and all of the biosorption-accessible holes were occupied by Zn (II) [22]. In isothermal studies, the concentrations of adsorbate were varied, whereas the adsorbent amount was kept constant. Figure 2A demonstrates that when the initial Zn (II) concentration was raised (20–300 mg/L), a rapid increase in the rate of biosorption was noticed up to the point when the pores of the biosorbent were completely filled with adsorbate, attaining an equilibrium state. Numerous isothermal models as mentioned below were applied to describe the adsorption properties.

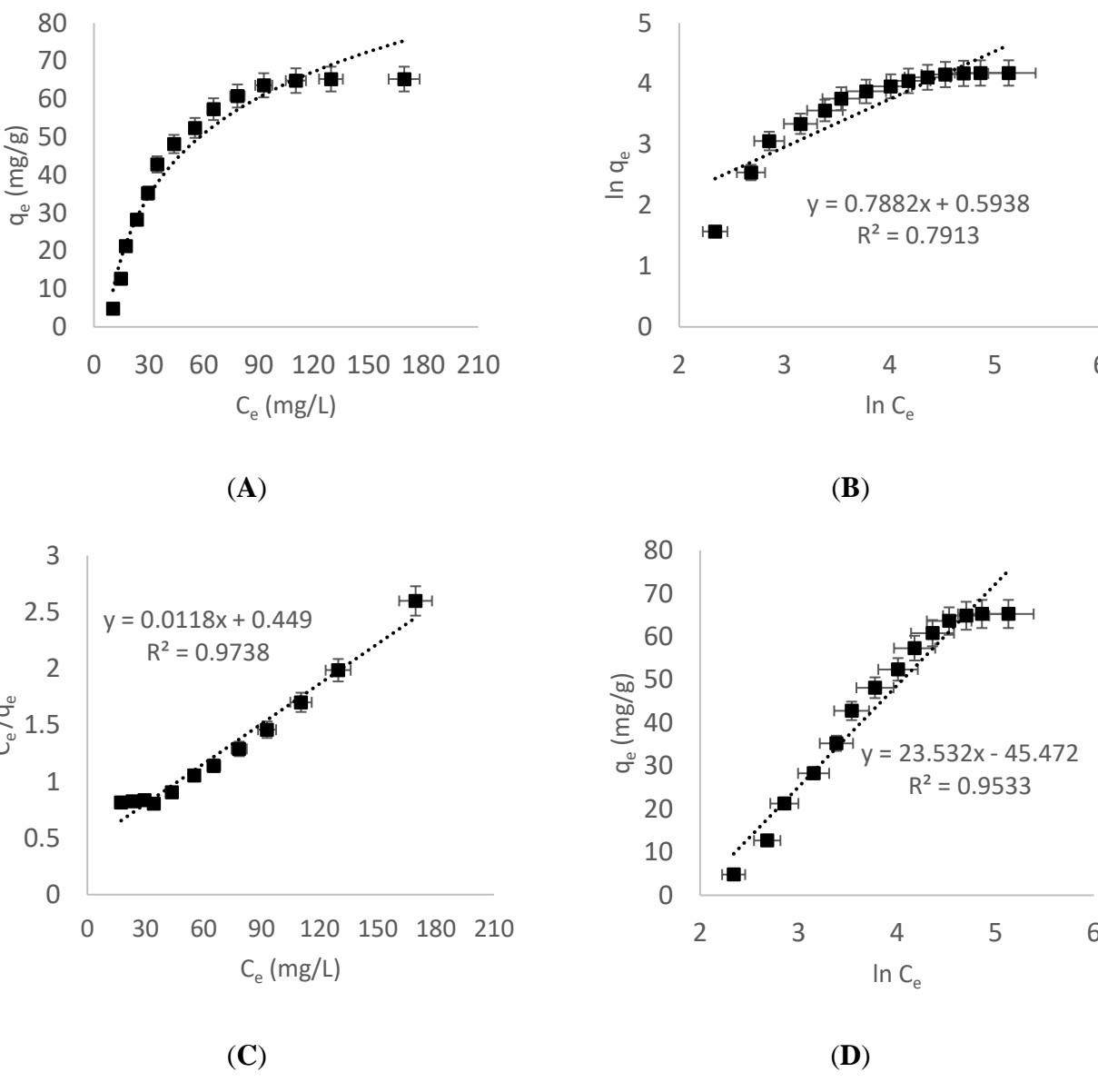

**Figure 2.** *Cont.*

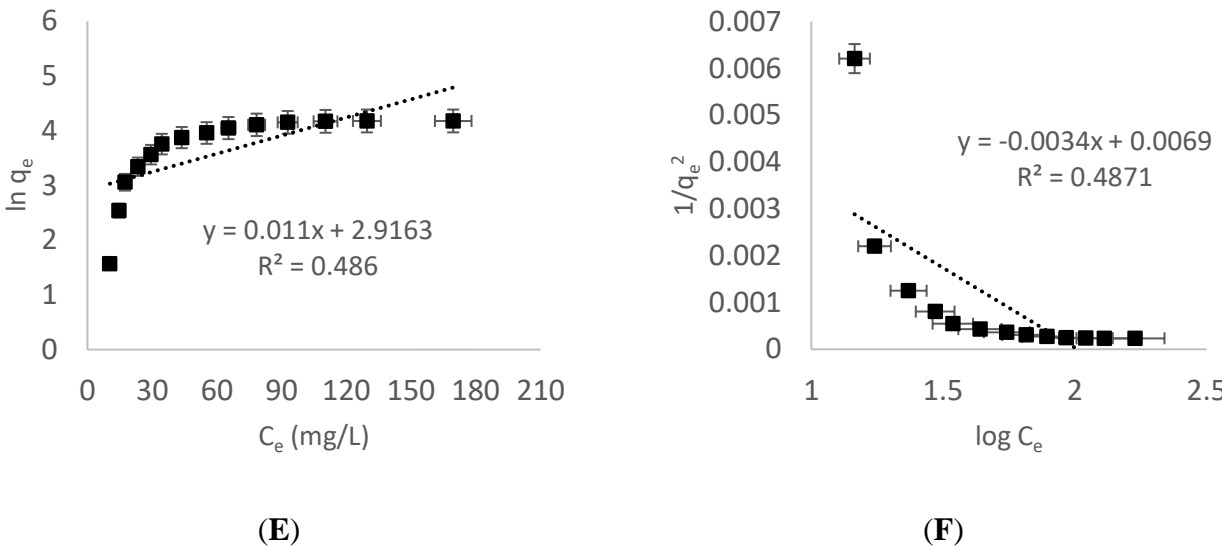

**(E)**                                                                 **(F)**

**Figure 2.** Isothermal study results: (**A**) influence of Zn (II) concentration, (**B**) Freundlich plot, (**C**) Langmuir isotherm, (**D**), Temkin isotherm, (**E**) Jovanovic plot, (**F**) Harkins–Jura plot.

### 3.2.1. Freundlich Isotherm

This model is used when one is dealing with heterogeneous adsorption systems [23]. Mathematically, it can be given as:

$$\ln q_e = \log K_F + \frac{1}{n}\ln C_e \tag{3}$$

where n is an empirical constant related to adsorption strength and $K_F$ is the Freundlich constant referring to the adsorption potential of an adsorbent. Figure 2B shows the isotherm of $\ln C_e$ vs. $\ln q_e$. The slopes and intercepts of the plot were used to derive the values of $K_F$ and n and are given in Table 2.

**Table 2.** Isothermal parameters for adsorption of Zn (II) on CMPVL.

| Adsorption Isotherm | Parameters | Values |
|---|---|---|
| Freundlich | $K_F$ (mg/g) | 1.810 |
| | n | 1.268 |
| | $R^2$ | 0.7913 |
| Langmuir | $q_{max}$ (mg/g) | 84.74 |
| | $K_L$ (L/mg) | 0.0262 |
| | $R^2$ | 0.9738 |
| Temkin | $\beta$ | 23.53 |
| | $\alpha$ | 6.889 |
| | b | 96.460 |
| | $R^2$ | 0.9533 |
| Jovanovich | $K_J$ (L/g) | 0.011 |
| | $q_{max}$ (mg/g) | 18.472 |
| | $R^2$ | 0.486 |
| Harkins–Jura | $A_H$ (g$^2$/L) | 0.294 |
| | $B_H$ (mg$^2$/L) | 2.029 |
| | $R^2$ | 0.4871 |

### 3.2.2. Langmuir Isotherm

This model illustrates the monolayer adsorption where it is assumed that the adsorbent has a limited number of surface-active sites of homogeneous distribution and that no interactions from further adsorbed molecules are anticipated after the adsorbent has interacted with particular pollutant molecules [24]. In equation form this model can be given as:

$$\frac{C_e}{q_e} = \frac{1}{K_L q_m} + \frac{C_e}{q_m} \tag{4}$$

Here $q_e$ shows the quantity of Zn (II) adsorbed at equilibrium, $C_e$ is the concentration of Zn (II) at equilibrium, $q_m$ represents the highest Zn (II) uptake capability by CMPVL, and $K_L$ is the Langmuir constant linked with biosorption energy. The graph of $C_e/q_e$ vs. $C_e$ shown in Figure 2C was utilized to estimate the quantities of $q_m$ and $K_L$, and these are listed in Table 2.

### 3.2.3. Temkin Adsorption Isotherm

This model suggests that surface coverage during the adsorption phase of the Zn (II) on the CMPVL was connected to the framework's free energy. The following mentioned equation can be utilized as the representative linear form of the model [25].

$$q_e = \beta \ln \alpha + \beta \ln C_e \tag{5}$$

Here, $\beta = RT/b$ where T = absolute temperature (K), b = adsorption heat constant, and R = general gas constant (8.314 J/mol.K). Using the slope and intercept of the graph shown in Figure 2D, the values of the constants were computed and are given in Table 2.

### 3.2.4. Jovanovic Isotherm Model

This model is used to represent the mechanical connections involved between the adsorbent and adsorbate. A mathematical representation of the model is given as below [26].

$$\ln q_e = \ln q_{max} - K_J C_e \tag{6}$$

Here, $C_e$ indicates the equilibrium adsorbate concentration, $q_e$ is the adsorbate amount adsorbed per unit of the adsorbent, and $q_{max}$ stands for the highest adsorption capacity of the adsorbent where $K_J$ is the Jovanovic constant. By plotting $\ln q_e$ versus $C_e$, the Jovanovic isotherm was obtained. Based on the slope and intercept of the curve, as given in Figure 2E, the computed data for $K_J$ and $q_{max}$ are listed in Table 2.

### 3.2.5. Harkins–Jura Isotherm

According to this isotherm, the adsorbent surface has a heterogeneous porous surface that allows multilayer adsorption. Mathematically, it is given as [27]:

$$\frac{1}{q_e^2} = \frac{B_H}{A_H} - \frac{1}{A_H} \log C_e \tag{7}$$

The plot of $1/q_e^2$ against $\log C_e$, as can be seen in Figure 2F, was used to check the isotherm's linearity, and the values of the constants $A_H$ and $B_H$ that are listed in Table 2.

The Langmuir model showed the highest $R^2 = 0.9738$ value among the different isothermal models employed, and thus it is the best model to accommodate the experimental data.

### 3.3. Kinetic Study

The kinetic investigation of Zn (II) adsorption utilizing CMPVL as a biosorbent was conducted at a concentration of 100 mg/L. For the first 30 min, a high rate of adsorption was seen continuing at a slow rate for 120 min. With time, the adsorption became consistent, and equilibrium was attained at 120 min. The adsorption process eventually slowed down after the rapid step, as initially more sites were available, which then reduced in number

as most of them were occupied by Zn from the solution [28] (Figure 3A). The adsorption kinetic parameters were computed using a number of kinetic models as described below.

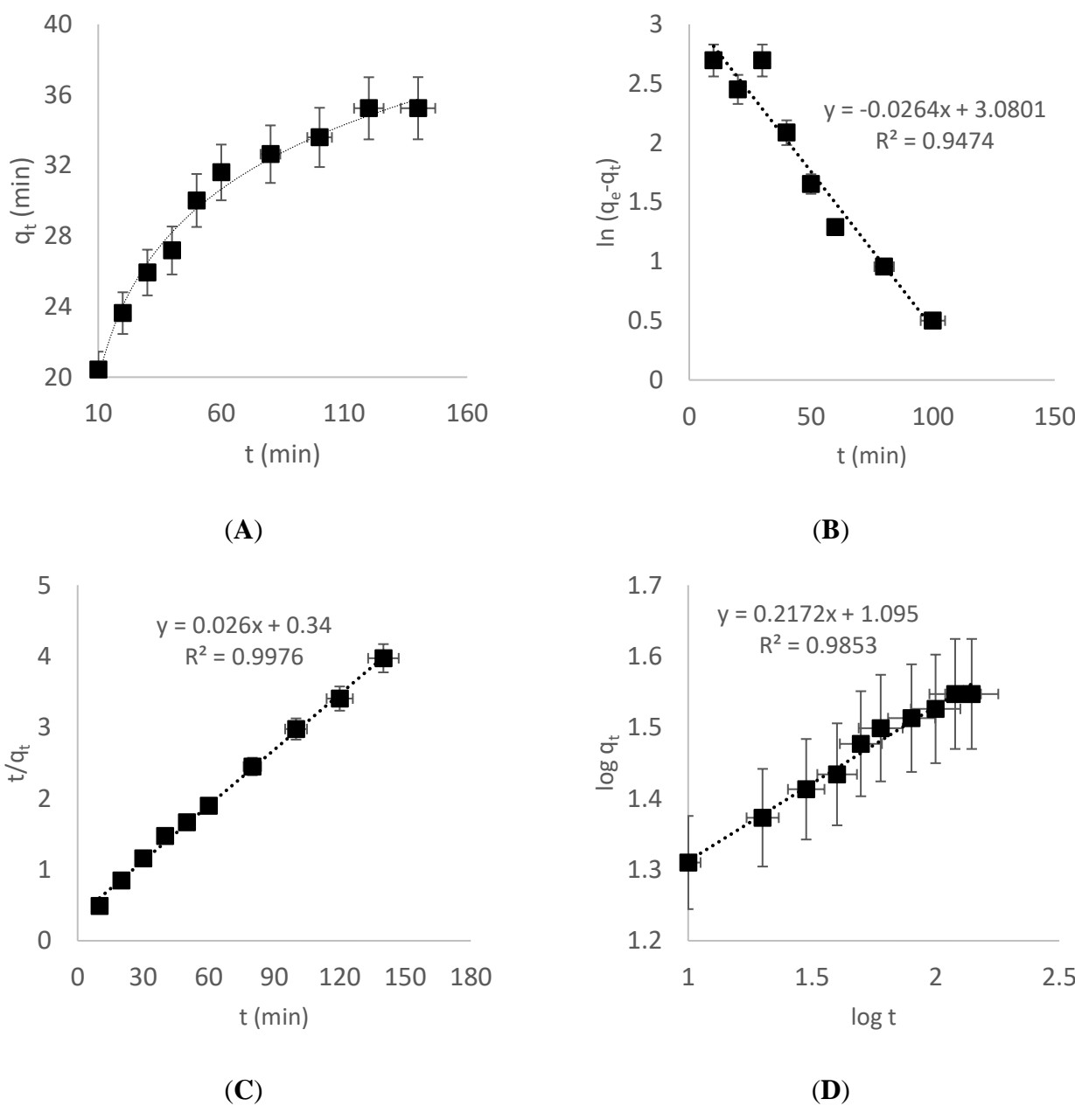

**Figure 3.** *Cont*.

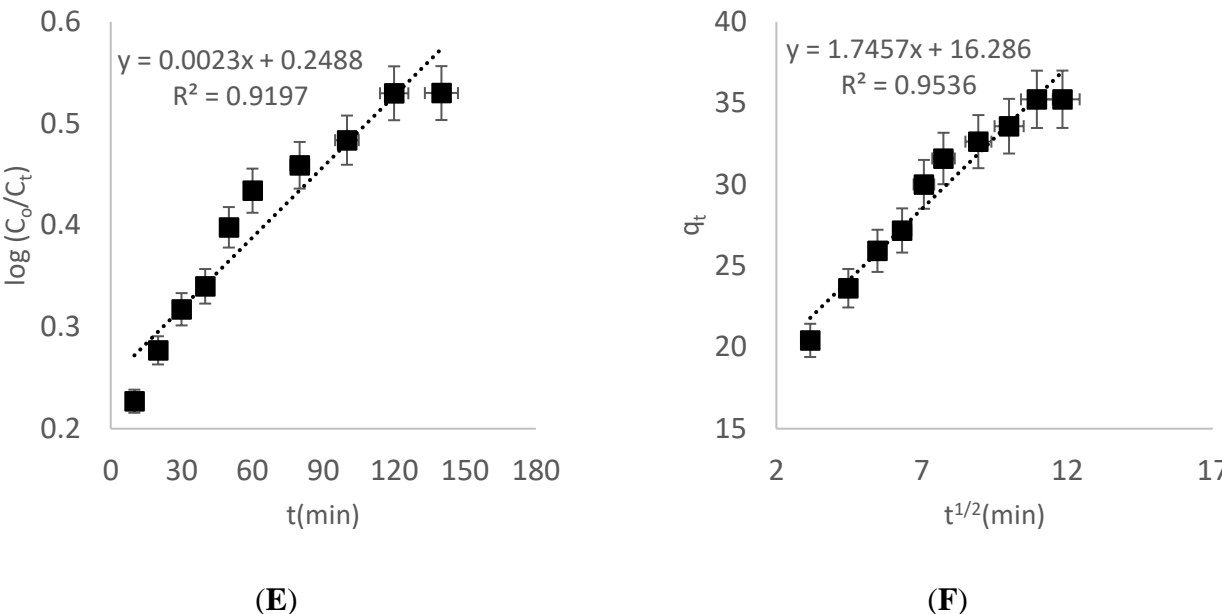

**(E)**                                **(F)**

**Figure 3.** Kinetic aspect of the process: (**A**) impact of contact time, (**B**) pseudo first order kinetic graph, (**C**) pseudo second order kinetic graph, (**D**) power function kinetic plot, (**E**) Natarajan–Khalaf kinetic plot, (**F**) intra particle kinetic plot.

### 3.3.1. Pseudo First Order Kinetic Model

The description of this isotherm can be mathematically given as follows [29]:

$$\ln(q_e - q_t) = \ln q_e - K_1 t \tag{8}$$

Here $K_1$ represents the first order rate constant, and $q_e$ and $q_t$ stand for the quantity of metal adsorbed at equilibrium and at time t, respectively. Table 3 shows the results of $K_1$ and $q_e$ computed from the plot's slope and intercept as shown in Figure 3B.

**Table 3.** Kinetic parameters of Zn (II) biosorption on CMPVL.

| Kinetic Model | Parameters | Values |
|---|---|---|
| Pseudo first order | $K_1$ (1/min)<br>$q_e$ (mg/g)<br>$R^2$ | −0.0264<br>21.760<br>0.9474 |
| Pseudo second order | $K_2$ (1/min)<br>$q_e$ (mg/g)<br>$R^2$ | 0.001988<br>38.46<br>0.9976 |
| Power function | $\alpha$<br>b<br>$R^2$ | 12.445<br>0.2172<br>0.9853 |
| Intra particle diffusion | $K_{diff}$ (mg/g min$^{1/2}$)<br>C<br>$R^2$ | 1.7457<br>16.286<br>0.9536 |
| Natarajan–Khalaf | $K_N$ (1/min)<br>$R^2$ | 0.00529<br>0.9197 |

### 3.3.2. Pseudo Second Order Kinetic Model

The expression of this model is illustrated as below [30].

$$\frac{t}{q_t} = \frac{1}{K_2 q_e^2} + \frac{t}{q_e} \tag{9}$$

where $K_2$ represents pseudo second order rate constant and its values were measured using the intercept of the plot given in Figure 3C, and their values are recorded in Table 3.

### 3.3.3. Power Function Kinetic Model

This isotherm expression can be given as follows [31]:

$$\log \; q_t \; = \; \log a \; + \; b \log t \tag{10}$$

The reaction rate constant values of b and a are given in Table 3, and were obtained using the slope and intercept of the graph between $\log q_t$ and $\log t$ as illustrated in Figure 3D.

### 3.3.4. Natarajan and Khalaf Kinetic Model

The mathematical form of the model is given as [32]:

$$\log\left(\frac{C_0}{C_t}\right) \; = \; \frac{K_N}{2.303} t \tag{11}$$

where initial and final concentrations are represented by $C_o$ and $C_t$, respectively. $K_N$ is a constant, which is deduced from the slope of the graph, as displayed in Figure 3E.

### 3.3.5. Intraparticle Kinetic Model

The model can be expressed as given below [33]:

$$q_t = K_{diff} \; t^{1/2} + C \tag{12}$$

Here $K_{diff}$ is the rate constant of this model. The parameter C corresponding to the thickness of the boundary layer was computed from the intercept of $q_t$ vs. $t^{1/2}$ of the chart displayed in Figure 3F, and their values are mentioned in Table 3.

### 3.4. pH Study

To measure the ideal pH at which the biosorption occurred, the biosorption of Zn (II) on CMPVL was investigated in the pH limit of 2–8 as demonstrated in Figure 4. The highest remediation of Zn (II) was observed at pH 6. At low pH, the adsorption capacity of CMPVL was limited due to the competition of Zn (II) and $H_3O^+$ ions. At lower pH, the concentration of $H_3O^+$ was high in the solution and occupied the active sites of the adsorbent resulting in the decreased adsorption of Zn (II) on CMPVL. At improved pH, the concentration of $H_3O^+$ steadily reduced and was eliminated from the material surface resulting in reduced competition between Zn (II) and $H_3O^+$, which allowed the metal ions to approach the active site of the biosorbent, resulting in increased biosorption of Zn(II) on CMPVL. Therefore, the optimum removal of 80.35% was seen at pH 6. When the pH was higher than 6, Zn (II) biosorption was low, and this was caused due to the precipitation of Zn (II) [34].

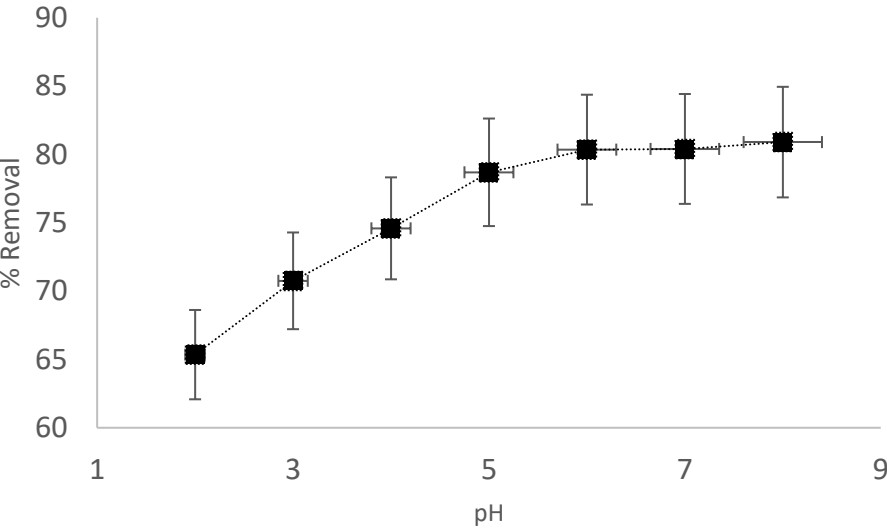

**Figure 4.** Effect of pH on biosorption process.

*3.5. Impact of the Biosorbent Dosage*

The effect of the biosorbent dosage (0.01–0.12 g) on the removal of Zn (II) from the solution was also investigated in this study. Figure 5 depicts the connection between the dosage of the adsorbent and the elimination of Zn (II). The elimination of Zn (II) was boosted by raising the dosage of the biosorbent from 0.01 to 0.12 g. At a higher biosorbent dose (0.1 g), the maximum remediation of Zn (II) was observed because of the increased surface area and the presence of more active sites [35,36]. Therefore, in the batch tests, 0.1 g was added to the solution of Zn (II), which was the optimum dose recorded with a removal efficiency of 67.18%.

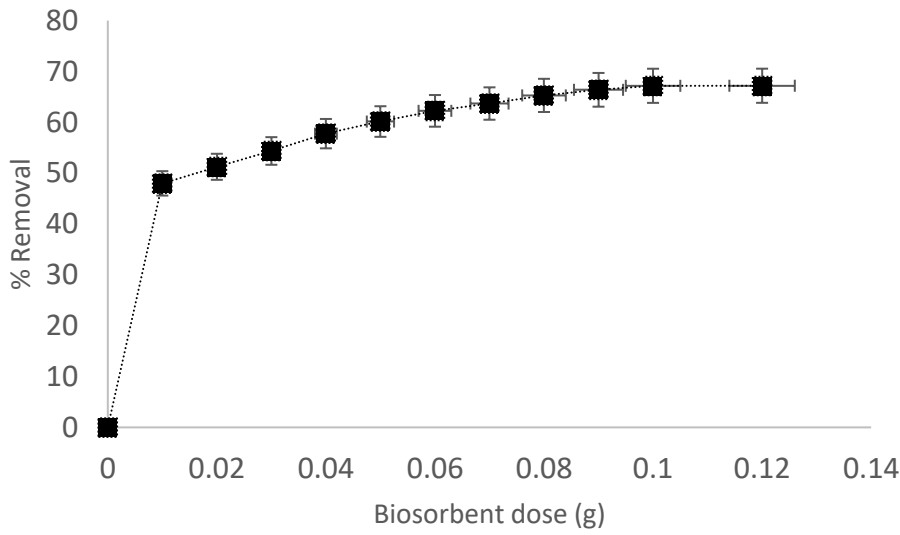

**Figure 5.** Effect of biosorbent dosage on biosorption process.

*3.6. Adsorption Thermodynamics*

From an industrial point of view, it is important to determine whether the adsorption process is exothermic or endothermic; therefore, a thermodynamics investigation was conducted to find out the favorable conditions for the process under study. Here the adsorption tests were performed at various temperatures (293, 303, 313, and 323 K). The $\Delta H°$ and $\Delta S°$, which represent the enthalpy and entropy change, respectively, were determined utilizing

the Van 't Hoff plot as given in Figure 6, and its numerical data are presented in Table 3. The mathematical description of the Van 't Hoff equation is described as follows [37,38]:

$$\ln K = \frac{\Delta S^\circ}{R} - \frac{\Delta H^\circ}{RT} \tag{13}$$

where $q_e$, $C_e$, R, and T are already mentioned above. $\Delta G^\circ$ stands for the Gibbs energy change that was computed by applying the formula as shown below.

$$\Delta G^\circ = \Delta H^\circ - T\Delta S^\circ \tag{14}$$

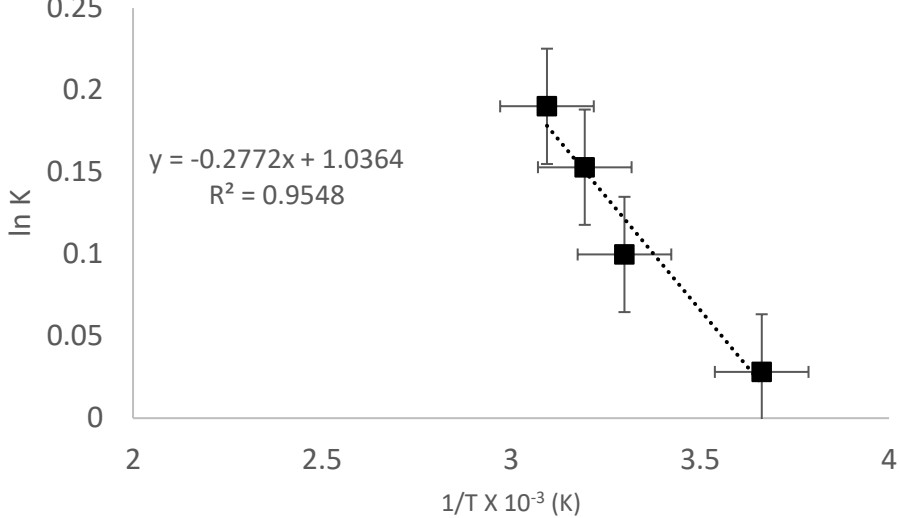

**Figure 6.** Thermodynamic plot of Zn (II) adsorption by CMPVL.

The calculated data of $\Delta G^\circ$ are displayed in Table 4, highlighting the favorable and spontaneous aspect of the process under study. The negative and positive values of $\Delta H^\circ$ and $\Delta S^\circ$, respectively, represents the exothermic and spontaneous nature of the process.

**Table 4.** Thermodynamic parameters of Zn (II) biosorption on CMPVL.

| Parameters Values | Values |
|---|---|
| $\Delta H^\circ$ (J/mol K) | −2.304 |
| $\Delta S^\circ$ (J/mol K) | 8.61 |
| T(K) | $\Delta G^\circ$ (KJ/mol) |
| 293 | −2.5.25 |
| 303 | −2.611 |
| 313 | −2.697 |
| 323 | −2.783 |

### 3.7. Comparative Study with the Literature

A comparative study was conducted between CMPVL and reported adsorbent adsorption capacities, and these are displayed in Table 5.

**Table 5.** Adsorption capacity of present adsorbent compared with those reported in literature.

| S. No. | Biosorbent | $q_{max}$ (mgg$^{-1}$) | References |
|:---:|:---:|:---:|:---:|
| 1 | Present work | 84.74 | |
| 2 | Modified Buxus sempervirens tree leaves | 21.2 | [39] |
| 3 | Nanostructured cedar leaf ash | 4.79 | [40] |
| 4 | Exhausted tea leaves | 79.76 | [41] |
| 5 | Bael tree leaf powder | 2.083 | [42] |

## 4. Conclusions

In this study, an efficient adsorbent was made from *Pteris vittata* leaves with natural affinity in the form of phytoremediation capability for the selected Zn metal. The leaves were chemically modified in order to enhance the biosorption process. By employing these modified leaves, Zn (II) was successfully removed from the steel mill effluents. The optimal experimental conditions established were: initial Zn (II) concentration = 100 mg/L, contact time = 2 h, pH = 6, biosorbent dosage = 0.1 g, and temperature = 30 °C. The most effective isotherm model among the tested ones was the Langmuir isotherm ($R^2$ = 0.9738). The pseudo second order kinetic model described the kinetics data well with an $R^2$ value of 0.9976, showing the chemical nature of the process. The thermodynamic investigation described that the Zn (II) biosorption on CMPVL was favorable and exothermic, as well as spontaneous. These findings suggest that CMPVL can be effectively used to eliminate Zn (II) from aqueous environments, and is a readily available, low-cost biosorbent. The CMPVL biosorbent utilized in this work needs some improvements in order to boost its capacity for biosorption further, and needs some other chemical treatments. It also needs to be tested for the biosorption of other pollutants.

**Author Contributions:** Experiments and write up of initial draft of manuscript, Q.K. and M.W.; Supervision, M.Z. and S.M.S.; Formal analysis, M.T., Y.K., A.W.K., E.A.A., R.U. and A.B.S.; Resources, E.A.A., R.U. and M.Z.; Review and proof reading, M.Z.; Project administration, M.Z. All authors have read and agreed to the published version of the manuscript.

**Funding:** The study was supported by Researchers Supporting Project Number RSP/2021/45, King Saud University Riyadh, Saudi Arabia.

**Institutional Review Board Statement:** Not applicable.

**Informed Consent Statement:** Not applicable.

**Data Availability Statement:** All the associated data is presented in this paper.

**Acknowledgments:** Authors are thankful to Researchers Supporting Project Number RSP/2021/45, King Saud University Riyadh, Saudi Arabia.

**Conflicts of Interest:** The authors declare no conflict of interest.

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
