# Peer review of "The Chemically Modified Leaves of Pteris vittata as Efficient Adsorbent for Zinc (II) Removal from Aqueous Solution"

_water, doi:10.3390/w14244039_

Round 1
Reviewer 1 Report
The authors present a new research that fits well with journal scope, the results of the tests are well described, figures are clear and easy to understand, moreover the discussion is clear and consistent with the achieved results. For these reasons, the paper can be considered for publication after reviewing it considering the following suggestions.
Page 2 lines 85-90: plant's leaves were cleaned to eliminate the dirt and dissolved pollutants, after the phytoremediation treatment which contaminants were found in Pteris vittate?
Page 3 lines 95-96: Has the removal of trapped metals been checked?
Page 4 lines 138-144: Please add the initial zinc concentration used in these tests.
Page 6 line 181: Replace Table 2 with Figure 1. Check caption of Figure 1 because letter D is present twice.
Page 6 lines 184-192: In these tests performed at different initial zinc concentration values, can you provide the range of removal efficiency? Does the removal efficiency decrease with increasing initial Zn concentration?
Page 7 line 237: replace A and B with AH and BH
Page 13 lines 296-297: How can this information be stated? is there an experimental proof? referring to a previous bibliographic study, it is better to use the conditional.
Figures 4 and 5: add a comment on the removal efficiency values achieved and if these values may depend on the initial zinc concentration (This paper might be helpful: “A comparison between Fe0/pumice and Fe0/lapillus mixtures in permeable reactive barriers”).
In the captions of Figures 4, 5 and 6 it is not necessary to insert the letter A
Paragraph 3.7: Specify if qmax value depends on the adsorbent dose used in the tests, if this dependence exists, it is necessary to include this parameter in all the studies summarized in table 4.
Author Response
Reviewer 1
The authors present a new research that fits well with journal scope, the results of the tests are well described, figures are clear and easy to understand, moreover the discussion is clear and consistent with the achieved results. For these reasons, the paper can be considered for publication after reviewing it considering the following suggestions.
- Thank you worthy reviewer for the encouraging remarks. We have tried out best to revise the paper according to your valuable suggestion.
Page 2 lines 85-90: plant's leaves were cleaned to eliminate the dirt and dissolved pollutants, after the phytoremediation treatment which contaminants were found in Pteris vittate?
- Worthy Reviewer, the steel industrial effluent were checked using AAS having the concentration of iron, Zinc and chromium. After the phytoremediation treatment the Pteris vittate plant leaves kept the Iron, Zinc and chromium in maximum quantity. The reference of the published paper is incorporated in the revised paper where the phytoremediation capability of the selected plant has been described.
Page 3 lines 95-96: Has the removal of trapped metals been checked?
- Worthy Reviewer, the removal of trapped metal has been checked per following detail: First HNO3 was utilized to remove trapped metal (Fe, Zn and Cr) from Pteris vittate leaves which was used as an adsorbent. The EDX analysis of unloaded adsorbent provides evidence that the Phyto remediated metals have been removed.
Page 4 lines 138-144: Please add the initial zinc concentration used in these tests.
- Worthy Reviewer, the initial concentration of Zn (II) solution used was (20-300 mg/L) and this concentration has been added in in the adsorption isothermal study. While for the determination of the influence of pH and biosorbent dose on adsorption study a fixed concentration of 100 mg/L were used. The information has been incorporated in the revised paper.
Page 6 line 181: Replace Table 2 with Figure 1. Check caption of Figure 1 because letter D is present twice.
- Worthy Reviewer, The Table 2 is replaced with Fig. 1 as well as the caption of Fig. 1 is checked and corrected. In Figure 1 the letter D was corrected accordingly.
Page 6 lines 184-192: In these tests performed at different initial zinc concentration values, can you provide the range of removal efficiency? Does the removal efficiency decrease with increasing initial Zn concentration?
- Worthy Reviewer, the initial metal concentration was wide ranging from 20-300 mg/L and from these concentrations the removal efficiency (qe) obtained was 4.8 to 65.25 mg/g. Once the active site of the adsorbent is occupied i.e fully saturated then further increase in concentration could not influence the removal efficiency as the adsorbent active sites becomes saturated with the metal.
Page 7 line 237: replace A and B with AH and BH
- Worthy Reviewer, the A and B was replaced with AH and BH.
Page 13 lines 296-297: How can this information be stated? is there an experimental proof? referring to a previous bibliographic study, it is better to use the conditional.
- Worthy reviewer, the reference of paper has been given in the revised paper. Also the statement has been rephrased.
Figures 4 and 5: add a comment on the removal efficiency values achieved and if these values may depend on the initial zinc concentration (This paper might be helpful: “A comparison between Fe0/pumice and Fe0/lapillus mixtures in permeable reactive barriers”)
- Worthy reviewer, the removal efficiency was put in pH study and impact biosorbent dose has been described in Figure 4 and 5. The section has been rephrased accordingly.
In the captions of Figures 4, 5 and 6 it is not necessary to insert the letter A
- Worthy reviewer, in the captions of Figures 4, 5 and 6 the letter A was removed.
Paragraph 3.7: Specify if qmax value depends on the adsorbent dose used in the tests, if this dependence exists, it is necessary to include this parameter in all the studies summarized in table 4.
- Worthy reviewer, the qmax value depends on the adsorbent nature and also different adsorbates have different qmax for the same adsorbate, That’s why we have compared qmax value off different adsorbents from isothermal parameters. Worthy reviewer we have performed our experiments under optimized condition therefore adsorbent dosage not effecting qmax values. In the adsorbent dosage effect studies it is evident that is dosage dependent however, as mentioned the experiment has been performed under optimized conditions.
Reviewer 2 Report
The authors should better explain the rationale behind the selection of Pteris vittate as biosorbent. What kind of test they performed with other plants? They claim that it has “remarkable phytoremediation propierties” but they don´t give any reference or test.
Why is the plant treated with Calcium chloride? Please explain, what “groups” are introduced during this step?
In line 115. A solution is not “manufactured”, it is “prepared”, or other suitable verb
Equation 1 and 2 are blurred, please improve the image
For the kinetic study, How the concentration of zinc was determined during the experiments with respect to time?
Section 3.1.1. There is no discussion on the FTIR results, the authors only list the peaks observed and assign possible bonds for each peak but there is no real information on the differences between the treated and loaded samples or the reasons (if any) why they may be different.
The quality of the EDS spectra are very poor, please improve them
Line 165-166. The authors say: “As shown in the SEM images, after adsorption Zn (II) had adhered to the surface of the biosorbent”. It is impossible to “see” ions from a SEM image.
Line 181. “Table 2” should be replaced by “Figure 1”
The TGA results are irrelevant as presented, the authors make no discussion or use of the information obtained from this technique
The data on the thermodynamic study are not reliable, the error bars are very broad and the R2=0.5 is very low. Either repeat the experiments or explain why they are obtained like this
The biggest issue with this paper is that there is no real comparison between the material treated chemically and the untreated material. All the relevant results (biosorption, kinetics, thermodynamics) are only shown for the treated material, therefore it is impossible to assess the improvement (if any) of the proposed material processing.
In the conclusions the authors claim: “The leaves were chemically modified in order to enhance the biosorption process”, but there is no proof that may lead to this conclusion, since there is no data on the unmodified material.
In the same section, the authors say: “The CMPVL biosorbent utilized in this work has to be improved in order to boost its capacity for biosorption.” But how this can be done?
Author Response
Reviewer 2
The authors should better explain the rationale behind the selection of Pteris vittate as biosorbent. What kind of test they performed with other plants? They claim that it has “remarkable phytoremediation propierties” but they don´t give any reference or test.
- Worthy reviewer, as in our recently published paper the nine wild plants were checked for their phytoremediation capability of Zn. Among these the Pteris vittate plant was determined the best one having optimum phytoremediation ability. The concentration of Zn was determined using Atomic absorption spectrophotometer. The following refence has been incorporated in the revised paper.
- Khan, Q.; Zahoor, M.; Salman S.M.; Wahab, M.; Bari, W. Phytoremediation of toxic heavy metals in polluted soils and water of Dargai District Malakand Khyber Pakhtunkhwa, Pakistan. Brazilian Journal of Biology, 2024, 84, e265278 | https://doi.org/10.1590/1519-6984.265278
Why is the plant treated with Calcium chloride? Please explain, what “groups” are introduced during this step?
- Worthy Reviewer, as far the organic functional groups are concerned, they could not be added with Calcium chloride treatment. It was done in order to add Ca+2, which performed a function in ion exchange for the removal of Zn(II).
In line 115. A solution is not “manufactured”, it is “prepared”, or other suitable verb
- Worthy Reviewer, thank you it is corrected using the suitable word prepared in the revised file.
Equation 1 and 2 are blurred, please improve the image
- Worthy Reviewer, Equation 1 and 2 were improved accordingly.
For the kinetic study, How the concentration of zinc was determined during the experiments with respect to time?
- Worthy Reviewer, the concentration of Zn(II) solution taken was 100mg/l and its adsorption was checked at definite interval of time in batch experiments. The concentration of Zn (II) was determined using atomic absorption spectrophotometer by performing the experiments in the range of 0 -140 min. After the experiment their concentration was checked as stated above. The information has been added into the revised file accordingly.
Section 3.1.1. There is no discussion on the FTIR results, the authors only list the peaks observed and assign possible bonds for each peak but there is no real information on the differences between the treated and loaded samples or the reasons (if any) why they may be different
- Worthy Reviewer, The FTIR spectra of loaded and unloaded samples was explained, in fact the whole section has been revised accordingly.
The quality of the EDS spectra are very poor, please improve them.
- Worthy Reviewer, These are machine generated graphs, however we have improved it by magnifying its size. Hopefully it will be ok now.
Line 165-166. The authors say: “As shown in the SEM images, after adsorption Zn (II) had adhered to the surface of the biosorbent”. It is impossible to “see” ions from a SEM image.
- Worthy Reviewer, you are right it is impossible to visualize such connectivity however, before adsorption the surface of adsorbent was porous, curved surface and bent edges which was best suited for adsorption. After adsorption these edges and pores were occupied by the adsorbate. So we may conclude that Zn occupied the pores and edges and adsorption has been taken place. The discussion about Zn adsorption was improved in the paper. It is probable reason rather than experimental evidence. At the same time we have rephrased the section to an acceptable statement.
Line 181. “Table 2” should be replaced by “Figure 1”
- Worthy Reviewer, the Table 2 was replaced by Fig. 1 accordingly.
The TGA results are irrelevant as presented, the authors make no discussion or use of the information obtained from this technique
- Worthy Reviewer, from TGA one can determines the mass change of a sample with respect to the temperature rise, and stability of given adsorbent can be determined using this technique. Moreover, it records the degradation of biosorbent and phases change occurred with respect to temperature. These information has been added into the revised paper.
The data on the thermodynamic study are not reliable, the error bars are very broad and the R2=0.5 is very low. Either repeat the experiments or explain why they are obtained like this
- Worthy Reviewer, the thermodynamic study and calculations were rechecked there was mistakes in sample numbering all the data were rearranged and the graph again plotted. Hopefully it will be ok now.
The biggest issue with this paper is that there is no real comparison between the material treated chemically and the untreated material. All the relevant results (biosorption, kinetics, thermodynamics) are only shown for the treated material, therefore it is impossible to assess the improvement (if any) of the proposed material processing.
- Worthy Reviewer, the untreated material already contained this metal in high amount. Using it in untreated form could lead to erroneous results. Therefore, it has not been used in the untreated form. The chemical modification is necessary as it enhances the adsorption capability as compared to untreated adsorbent by introducing some group which help in the ion exchange process. Also, nitric acid treatment remove the already present metals.
In the conclusions the authors claim: “The leaves were chemically modified in order to enhance the biosorption process”, but there is no proof that may lead to this conclusion, since there is no data on the unmodified material.
- Worthy Reviewer, the plant is taken from a polluted area where the selected plant already has high phytoremediation capability of Zn. See our published paper. The EDX analysis shows that there is no attached Zn in the chemically treated sorbent whereas our published result shows that high quantity of zinc in the leaves in present which is a clear proof. Please see our published paper available online.
In the same section, the authors say: “The CMPVL biosorbent utilized in this work has to be improved in order to boost its capacity for biosorption.” But how this can be done?
Worthy Reviewer, The improvement can be done by chemical modification of biosorbent utilizing some other chemical to further boost the recorded capacity in the present study. It is future perspective of the study as we want 100% efficiency. From such type of statement other scientists gets an insight to further explore them.
Reviewer 3 Report
This article reports the performance of new adsorbent prepared form leaves, for efficient removal of zinc from water. In my opinion, this manuscript contains sufficient scientific evidence and clarity on the adsorption process.
However, the authors should include the following points,
· The SEM images of spent adsorbent should be included in order to understand its morphology after adsorption.
· Specific Surface area (BET) of the adsorbent, should be reported.
· The insights into isotherm studies should be mentioned depending on the nature of the adsorption process.
Author Response
Reviewer 3
This article reports the performance of new adsorbent prepared form leaves, for efficient removal of zinc from water. In my opinion, this manuscript contains sufficient scientific evidence and clarity on the adsorption process.
- Thank you worthy reviewer, for the positive input.
However, the authors should include the following points,
- The SEM images of spent adsorbent should be included in order to understand its morphology after adsorption.
- Worthy Reviewer, The SEM images of loaded biosorbent is presented in the form of fig. 1D accordingly.
- Specific Surface area (BET) of the adsorbent, should be reported.
- Worthy Reviewer, Surface area (BET) of the adsorbent is presented in table 1.
- The insights into isotherm studies should be mentioned depending on the nature of the adsorption process.
- Worthy Reviewer, from isotherm studies different adsorption parameters have been obtained which give an insight into adsorption process. According to kinetics investigation biosorption of Zn (II) on CMPVL follow pseudo second order kinetics pointing towards the chemical nature of the process and added in the conclusion section of paper. The required detail in the respective sections has been incorporated accordingly.
Reviewer 4 Report
1-Throughout the manuscript, the authors must check that the numerical value they report is separated from the corresponding unit of measure.
2- What is the novelty of the research presented by the authors? This is important to be established in the introduction of the document, because there are already, to date, many articles published on the removal of metals by adsorption or biosorption, even with synthetic materials. Possible to use in environmental applications where the amounts of adsorbent required are large.
3- In the introductory section on line 60-61, the font size is smaller than the rest of the font used in the manuscript, also in page 2 lines 90-91.
4 authors should write the units of measurement according to the international system of units, for example "h" instead of "hrs". review this detail throughout the document.
5- on page 3 line 105, they should correct the units of the Infrared wavelength because it says 4000 1/cm instead of cm-1, where the number 1 should be written as exponential or superscript.
6- in various places in the manuscript, in the text, there are different letter sizes. Please fix this throughout the document, for example on page 3 at lines 108-109, I also get this error.
7- on line 122 page 3, you must enter the units of measurement for Ci and Cf.
8- In adsorption studies, the adsorption kinetics are first performed and written, and then the isotherm. Correct this order in the manuscript because it is reversed.
9- The description of the pre-treatment to which the plant was subjected is confusing and not clear, for this reason I consider it should be modified and more explicit. For example, what is the objective of activating with CaCl2, perhaps the Ca ions could not be adsorbed on the surface of the material? How do you confirm that there is no previous adsorption of this metal on the surface of the plant? In that case it should include the characterization for the material without pretreatment and after adsorption as well.
10- I do not see the relevance of the thermal analysis since they will not work with such high temperatures in the adsorption process. However, considering that there are other techniques that provide more information for this study, one is the physisorption of N2 to determine textural properties: surface area, type and pore size) and the other is the determination of the pH at the point of zero load, which also provides information on the surface of the material and associate it with the pH studies that the authors mention.
11- the units of measure for Freundlich's constant Kf and Langmuir's constant Kl are missing.
12- the authors must include the units of measurement of the constants of the models that are not dimensionless.
13- in the application of mathematical models there is an error and it is that when linear models are used, the data used is the one that corresponds only to the linear region. Maybe that's why the settings are so different and it's something you should check. In any case, I suggest the authors to apply the non-linear models instead of the linear ones for a better accuracy of the results obtained and the corresponding discussion.
14- I consider that the authors should mention the physical meaning of the kinetic models since they do not do so, that is, what type of adsorption they describe, and the latter should also apply the nonlinear models. Nor should they include in the definition of each model variable, include the units of measures in which they are expressed.
15- in the kinetic graphs, the x-axis is represented as time, but instead of the complete word, they must write the time symbol, that is, t(min).
16-t1/2, also has units in which they are expressed and the authors must include it in the graph they report (Figure 3f).
17- I consider that the article addresses a strong environmental problem. However, they do not establish its novelty and what makes it different from other articles published on the same subject. You have several issues in the document to check and discuss thoroughly.
Author Response
Reviewer 4
1-Throughout the manuscript, the authors must check that the numerical value they report is separated from the corresponding unit of measure.
Worthy Reviewer, the instruction were followed accordingly. Thank you for the suggestion.
2- What is the novelty of the research presented by the authors? This is important to be established in the introduction of the document, because there are already, to date, many articles published on the removal of metals by adsorption or biosorption, even with synthetic materials. Possible to use in environmental applications where the amounts of adsorbent required are large.
Worthy Reviewer, the novelty of the research was discussed in the introduction. Here we have used the idea of phytoremediation and the best phyto-remediating plant has selected in the design of efficient adsorbent to remove Zn from steel mill effluent causing water pollution of Dargai Malakand Pakistan. The novelty statement was incorporated accordingly as: In the current work, we have combined the idea of phytoremediation in the design of an efficient adsorbent after deciding the best plant (Pteris vittate) that was capable of remediating zinc whose leaves were then chemically modified to get an efficient biosorbent.
3- In the introductory section on line 60-61, the font size is smaller than the rest of the font used in the manuscript, also in page 2 lines 90-91.
- Worthy Reviewer, the correction was made accordingly.
4 authors should write the units of measurement according to the international system of units, for example "h" instead of "hrs". review this detail throughout the document.
- Worthy reviewer, The units hrs has been replaced by h in the entire manuscript.
5- on page 3 line 105, they should correct the units of the Infrared wavelength because it says 4000 1/cm instead of cm-1, where the number 1 should be written as exponential or superscript.
- Worthy reviewer, the units has been corrected in the revised paper.
6- in various places in the manuscript, in the text, there are different letter sizes. Please fix this throughout the document, for example on page 3 at lines 108-109, I also get this error.
- Worthy Reviewer, the corrections were made to fix the problem throughout the document.
7- on line 122 page 3, you must enter the units of measurement for Ci and Cf.
- Worthy reviewer, the units of Ci and Cf was inserted in the above mentioned page
8- In adsorption studies, the adsorption kinetics are first performed and written, and then the isotherm. Correct this order in the manuscript because it is reversed.
- Worthy reviewer, yes you are as from kinetics studies we decide the equilibrium time which important and save our precious time rather than performing experiments for longer time. In literature in some papers kinetics is presented first then the isotherms however, in dominant literature the isotherms are first presented. If we revise the sequence at this stage, we will have to revise the figures and tables, also the references therefore we left them unchanged. If worthy reviewer still wish then we will revise the sequence.
9- The description of the pre-treatment to which the plant was subjected is confusing and not clear, for this reason I consider it should be modified and more explicit. For example, what is the objective of activating with CaCl2, perhaps the Ca ions could not be adsorbed on the surface of the material? How do you confirm that there is no previous adsorption of this metal on the surface of the plant? In that case it should include the characterization for the material without pretreatment and after adsorption as well.
- Worthy reviewer, pretreatment is mandatory as the plant is collected from the steel mill drains where the plant have remediated high amount of Zn from water and soil. The material is already contained high amount of Zn thus if used as such would have less adsorption capacity. Nitric acid removed the metals as evidenced from EDX of treated sample. It is further confirmed by EDX that the treated sample possess Ca and after adsorption of Zn the loaded sample showed peak of Zn and is described in the EDX section. The CaCl2 activation is carried out to enhance the ion exchange process. Here Ca exchange for Zin that is why it is needed to be there on the surface of the sorbent.
10- I do not see the relevance of the thermal analysis since they will not work with such high temperatures in the adsorption process. However, considering that there are other techniques that provide more information for this study, one is the physisorption of N2 to determine textural properties: surface area, type and pore size) and the other is the determination of the pH at the point of zero load, which also provides information on the surface of the material and associate it with the pH studies that the authors mention.
- Worthy Reviewer, the thermal analysis shows the stability of the material at high temperature if used. You are here it do not have any relevance but if some one uses high temp in future the findings will then be useful.
11- the units of measure for Freundlich's constant Kf and Langmuir's constant Kl are missing.
- Worthy Reviewer, their units are shown in table 1 accordingly.
12- the authors must include the units of measurement of the constants of the models that are not dimensionless.
- Worthy Reviewer, the units of different constants of models of the constants are given in the given table
13- in the application of mathematical models there is an error and it is that when linear models are used, the data used is the one that corresponds only to the linear region. Maybe that's why the settings are so different and it's something you should check. In any case, I suggest the authors to apply the non-linear models instead of the linear ones for a better accuracy of the results obtained and the corresponding discussion.
- Worthy reviewer, being not expert in mathematics it is difficult for us to apply such non linear consideration. However, in our future study we will try to learn them apply.
14- I consider that the authors should mention the physical meaning of the kinetic models since they do not do so, that is, what type of adsorption they describe, and the latter should also apply the nonlinear models. Nor should they include in the definition of each model variable, include the units of measures in which they are expressed.
Worthy Reviewer, as the data fits well pseudo second order kinetics model representing the chemical nature of the process. This point is added and written in conclusion section. For non linear model see explanation provided above.
15- in the kinetic graphs, the x-axis is represented as time, but instead of the complete word, they must write the time symbol, that is, t(min).
- Worthy Reviewer, it was corrected and added in graph accordingly.
16-t1/2, also has units in which they are expressed and the authors must include it in the graph they report (Figure 3f).
- The correction was made and added in the graph.
17- I consider that the article addresses a strong environmental problem. However, they do not establish its novelty and what makes it different from other articles published on the same subject. You have several issues in the document to check and discuss thoroughly.
- Worthy Reviewer, here the removal of Zn is checked by different plants and the plant having high phyremediation capability, leaves was changed into an efficient biosorbent. Here the idea of phytoremediation is utilized to design an efficient adsorbent which is a novel work and have not used by any researcher for the removal of Zn from steel mill effluent.
- Worthy Reviewer, the novelty of the research was discussed in the introduction. Here we have used the idea of phytoremediation and the best phyto-remediating plant has selected in the design of efficient adsorbent to remove Zn from steel mill effluent causing water pollution of Dargai Malakand Pakistan. The novelty statement was incorporated accordingly as: In the current work, we have combined the idea of phytoremediation in the design of an efficient adsorbent after deciding the best plant (Pteris vittate) that was capable of remediating zinc whose leaves were then chemically modified to get an efficient biosorbent.
Round 2
Reviewer 2 Report
The issue with the use of Calcuim chloride is still not explained. The text says “The activation process was necessary in order to introduce a certain group to the biosorbent that could help in Zn ion exchange from solution to biosorbent”. What is THIS CERTAIN GROUP?
The explanation for the differences in morphologies with the SEM images is completely wrong. The pores that the authors mention from the SEM images are micrometric, while the Zn ions measure ~0.19 nm. It is impossible to fill such micrometric pores with Zn ions. The differences in morphology by SEM are usually misleading because we can only see a very small portion of the sample, and it is not representative most of the times, especially for biological samples.
Regarding the TGA analysis, there still no reason to present this graph. I agree that using TGA we can monitor the mass change against temperature. However, what is the relevance of this technique in the Zn removal? What can we conclude from these results that is relevant towards the objective and hypothesis of this study?
Author Response
Reviewer 2
The issue with the use of Calcuim chloride is still not explained. The text says “The activation process was necessary in order to introduce a certain group to the biosorbent that could help in Zn ion exchange from solution to biosorbent”. What is THIS CERTAIN GROUP?
- Worthy Reviewer, after acid treatment there is need to neutralize the surface of biosorbent so as to offer suitable surface for cation adsorption. The deposited calcium chloride not only neutralizes the surface but also provides counter ion for zinc exchange. Activated carbon can also be produced via chemical activation of charcoal. This process involves soaking charcoal in a strong dehydrating agent such as calcium chloride (CaCl2) or zinc chloride (ZnCl2) after it has been carbonized at high temp. Various explanation has been provided in literature. In one paper it has been written that the CaCl2 modify the surface of adsorbent as CaCl2 treated biosorbent have shown maximum removal of a given pollutant as compared to untreated adsorbentst. After modification with CaCl2 the biosorbent surface becomes uniform, cylindrical, shrunken, and ruptured thus providing a suitable surface for adsorption as described in the following cited paper. In other studies, it has been used to provide positive surface for sorption which can also act as counter ion to be exchanged for Zinc ion (Feris et al 2005 and Salman et al 2020). The section was corrected per detail described above.
- In one study it has been used as per detail: Calcium chloride has been used as a mild microporogen for carbon materials. CaCl2 activation is advantageous over potassium hydroxide due to its lower relative cost, non-corrosiveness, and favorable stabilizing interactions with nitrogen functionalities similar to those present in carbon nitride.
- Also, in one study it has been pointed out that Calcium chloride pretreatment significantly changed the surface structure of carbonized fibers (https://journals.plos.org/plosone/article?id=10.1371/journal.pone.0212886)
- Reference: Sirilamduan, C., Umpuch, C., &Kaewsarn, P., Removal of copper from aqueous solutions by adsorption using modify Zalaccaedulis peel modify. Songklanakarin Journal of Science & Technology. 2011, 33(6).
- Féris, L. A., Misra, M. and Smith, R. W., Removal of nitrate using modified activated carbon – Part I. Brazilian Journal of Material Science and Engineering, 7, p. 61-69 (2005).
- Salman et al Pak. j. sci. ind. res. Ser. A: phys. sci. 2020 63A(1) 18-29.
- Burrage offered an explanation of the mechanism of activation of charcoal from vegetable matter based on the differential solvent action of the activating agents on cellulose and lignin. According to him, it is the cellulose or the charcoal formed from it which is being acted upon during activation whereas the lignin or the resultant charcoal is left untouched. This would cause pitting and it is this which activates charcoal. The active points are the carbonized remnants of cellulose (consisting of unoriented carbon atoms) attached to the walls of these activation pores.
- J. Burrage, Trans. Faraday Soc., 29, 445 (1933).
- Kandilarow stressed the importance of the formation of unoriented carbon atoms in the process of activation. But various modifications of the same process of activation are often found to alter the adsorption characteristics of charcoal for different types of adsorbates in different ways.
- G. Kandilarow, Kolloid-Beihe/te, 48, 1 (1938).
The explanation for the differences in morphologies with the SEM images is completely wrong. The pores that the authors mention from the SEM images are micrometric, while the Zn ions measure ~0.19 nm. It is impossible to fill such micrometric pores with Zn ions. The differences in morphology by SEM are usually misleading because we can only see a very small portion of the sample, and it is not representative most of the times, especially for biological samples.
- Worthy reviewer, the SEM images were replaced by high magnification SEM images to see clear differences. You are right worthy reviewer; zinc ions are quite small to fill the micrometric pores. The section was rephrased accordingly.
Regarding the TGA analysis, there still no reason to present this graph. I agree that using TGA we can monitor the mass change against temperature. However, what is the relevance of this technique in the Zn removal? What can we conclude from these results that is relevant towards the objective and hypothesis of this study?
- Worthy reviewer, the thermal properties of adsorbents were studied, in order to conclude about the stability of adsorbent if used at high temperature. In almost all adsorption studies, no one has used the adsorbents at high temperatures as almost all studies are carried out at room temperature but still, they have included such analysis. There are also there is no relevance as you pointed out but still included them to predict that if hot conditions are used then up to which temperature the adsorbent could be effectively used. Here also, we have included it find its stability at high temperature. As the steel industrial effluent having comparatively high temperature and if in situ it is used, how it behaves in that case.
Reviewer 4 Report
-incorporate in the introduction novel information about the bioadsorbent that is proposed in the work.
- in the materials and methods section should include the description of the determination of the surface area.
- authors must report pore size, shape, and distribution.
- The article has been correctly corrected, however, the novelty is not observed in this document and it is very similar to others that have been reported. I believe that new information should be incorporated that highlights the importance of the article
Author Response
Reviewer 4
-incorporate in the introduction novel information about the bioadsorbent that is proposed in the work.
Worthy reviewer the novel information’s about the bioadsorbent were included in the introduction section.
- in the materials and methods section should include the description of the determination of the surface area.
- Worthy Reviewer, In the materials and methods section the description of surface area and pore parameters determination were included.
- authors must report pore size, shape, and distribution.
- Worthy Reviewer, The pore volume and pore diameter is shown in the form of Table 1. Table 1 modified. And the pore shapes were and their distribution were irregular as it is clear from SEM images.
- The article has been correctly corrected, however, the novelty is not observed in this document and it is very similar to others that have been reported. I believe that new information should be incorporated that highlights the importance of the article.
- Worthy reviewer, The importance and novelty of this work is that the idea of phytoremediation and adsorption were combined, First the nine plants were checked for their high phytoremediation ability and then the high phytoremeditaion plant (Pteris vittata) were chemically modified and utilized in adsorption techniques, Moreover the said plant was not tested till now for zinc removal.
Round 3
Reviewer 4 Report
Line 154 page 4, the authors should writte "two hour" instead "two h"
line 162 and 163 page 4, the number and unit of measurement should be written separated. review this detail throughout the manuscript
The authors in Surface area and pore volume section, should add the interpretation about pores sizes.
authors should fit their isotherms and kinetics with nonlinear rather than linear models. What they represent is wrong because the statistical parameter (R2) is extremely low and that happens because they are applying the linear models incorrectly. However, I indicate that they should adjust it to the non-linear models instead of the linear ones, since the non-linear ones are more exact and give results more in line with the physical meaning of the process.
use the same way of writing the units of measurement throughout the document, for example in table 3 they indicate: K1 (1/min) instead of K1 (min-1, the latter as a superscript). Check this detail throughout The manuscript.
Author Response
Reviewer 4 comments
Line 154 page 4, the authors should writte "two hour" instead "two h"
- Worthy Reviewer, The word two h was replaced by two hour in entire manuscript.
line 162 and 163 page 4, the number and unit of measurement should be written separated. review this detail throughout the manuscript.
- Worthy Reviewer, The number and unit of measurement was separated in the entire manuscript.
The authors in Surface area and pore volume section, should add the interpretation about pores sizes.
- Worthy Reviewer the interpretation about pore pores size has been provided in the respective table.
authors should fit their isotherms and kinetics with nonlinear rather than linear models. What they represent is wrong because the statistical parameter (R2) is extremely low and that happens because they are applying the linear models incorrectly. However, I indicate that they should adjust it to the non-linear models instead of the linear ones, since the non-linear ones are more exact and give results more in line with the physical meaning of the process.
- Worthy reviewer, this paper is from a PhD project where the claimed study includes linear model usage. Converting them to nonlinear formats will create issues in award of degree. Hope the worthy reviewer will have understand our compulsions. In future study we will incorporate/follow this approach.
use the same way of writing the units of measurement throughout the document, for example in table 3 they indicate: K1 (1/min) instead of K1 (min-1, the latter as a superscript). Check this detail throughout The manuscript.
- Worthy Reviewer, In the entire manuscript for all units we have followed the pattern of 1/min except in FTIR discussion where the pattern is in the form of superscript which was suggested by worthy Reviewer 1, That always write the unit of FTIR in the form of cm—1